# Experimental and Numerical Study of the Influence of Pre-Existing Impact Damage on the Low-Velocity Impact Response of CFRP Panels

**DOI:** 10.3390/ma16030914

**Published:** 2023-01-18

**Authors:** Mohammad Rezasefat, Alessio Beligni, Claudio Sbarufatti, Sandro Campos Amico, Andrea Manes

**Affiliations:** 1Politecnico di Milano, Dipartimento di Meccanica, Via La Masa 1, 20156 Milano, Italy; 2PPGE3M, Federal University of Rio Grande do Sul, Porto Alegre 91501-970, Brazil

**Keywords:** CFRP, Puck failure criterion, low-velocity impact, pre-existing damage, numerical simulation

## Abstract

This paper presents an experimental and numerical investigation on the influence of pre-existing impact damage on the low-velocity impact response of Carbon Fiber Reinforced Polymer (CFRP). A continuum damage mechanics-based material model was developed by defining a user-defined material model in Abaqus/Explicit. The model employed the action plane strength of Puck for the damage initiation criterion together with a strain-based progressive damage model. Initial finite element simulations at the single-element level demonstrated the validity and capability of the damage model. More complex models were used to simulate tensile specimens, coupon specimens, and skin panels subjected to low-velocity impacts, being validated against experimental data at each stage. The effect of non-central impact location showed higher impact peak forces and bigger damage areas for impacts closer to panel boundaries. The presence of pre-existing damage close to the impact region leading to interfering delamination areas produced severe changes in the mechanical response, lowering the impact resistance on the panel for the second impact, while for non-interfering impacts, the results of the second impact were similar to the impact of a pristine specimen.

## 1. Introduction

The response of composite materials to out-of-plane impact loads during their service life is of concern in many applications. The low-velocity impact may cause interlaminar and intralaminar failure modes, such as matrix or fiber failure in tension and compression, crushing, and delamination. Matrix failure and delamination are particularly important since they can be invisible to the naked eye and still cause severe material degradation [1,2,3]. The low-velocity impact response of laminates has been the focus of many recent studies, especially related to a single hit [4,5,6,7,8]. Although repeated impacts are not uncommon in real-world applications of composites [9,10], there are not many studies in the literature addressing that, either numerically- or experimentally-based, as most focus on multiple hits at the same point [11]. The studies on the repeated impact behavior of composites mostly concern repeated impact at the same location by considering different impactor or target features, such as impactor geometry [12,13,14] and mass [15], reinforcement type [16,17,18,19], laminate thickness [17,20], and stacking sequence [21,22]. Atas et al. [23] investigated the effect of thickness on the repeated low-velocity impact response of E-glass/epoxy composites reporting the energy profiling diagram and the perforation limit. Liao et al. [13] performed an experimental study on damage accumulation of a repeated low-velocity impact in CFRP laminates considering different diameters of the hemispherical impactor. They presented a damage index based on stiffness reduction caused by repeated impacts which correlated well with the damage observed by a C-scan ultrasound.

Numerical simulations have proven to be an effective tool for the prediction of repeated low-velocity impacts on composites, despite the complexity of the loading condition, the constitutive model, and the multi-failure modes of composites [24]. For instance, Zhou et al. [25] developed a numerical model based on Hashin failure criteria to study CFRP laminates and observed a reduction in energy absorption for the second impact. Additionally, in a previous publication of the authors [26], the effect of repeated impacts on the initiation and propagation of interlaminar and intralaminar damage in CFRP laminates with each impact was investigated. By developing a model based on the Puck failure criterion with a constant fracture angle, the accumulation of damage was assessed. The CFRP laminate was impacted five times at the center, and the presence of damage from the first impact led to an increase in peak impact force [27]. High-fidelity macro-mechanical models have been able to predict intralaminar and interlaminar damage and permanent indentation [4], and also matrix damage and delamination [28,29]. The Puck inter-fiber failure criterion has been accurately applied to CFRP [29,30], as in the work of [31], who developed a strain-based progressive damage model with IFF of Puck to simulate laminates subjected to low-velocity impacts. Progressive delamination in a woven composite subjected to repeated impacts was simulated in [32] using a cohesive zone model, showing that delamination propagates rapidly after the first impact.

Regarding the repeated impacts at different locations, only a few experimental studies can be found [33,34], despite the practical engineering interest. Nassir et al. [35] studied multiple impacts in plain-weave glass fiber/epoxy laminates and showed that the critical force for the generation of damage does not change for a second hit at a different location. In addition, Liao et al. [36] investigated double impacts in CFRP laminates and used four different distances for the sequential impacts, concluding that the interference between the first and the second impact damage areas has a pivotal role in the response of the second impact. For impacts with interfering damaged areas, lower energy absorption was observed in the second impact, while the opposite phenomenon was observed for non-interfering damaged areas. More recently, Huang et al. [37] showed that for repeated impacts at different locations, the maximum displacement better characterizes the damage interference compared to the bending stiffness.

Considering the cited work, it is clear that further studies are required, especially numerically, to understand the complex damage behavior of composites subjected to multiple impacts. Additionally, there is a lack of studies on repeated impacts at different locations even though they are more likely to occur compared to repeated impacts at the same location. Thus, the aim of the current study is to numerically investigate the effect of the damage caused by a low-velocity impact on the structural integrity and impact performance of CFRP panels subjected to a subsequent hit at different locations, focusing on their interaction.

## 2. Experimental Study

The whole experimental campaign included three different levels of experimental tests on CFRP specimens obtained from unidirectional epoxy-based prepregs (CYCOM^®^ 977-2-34%-24K IMS-196-T1, Solvay, Brussels, Belgium), namely: (i) tensile and shear tests, (ii) low-velocity impact tests on coupon specimens (150 mm × 100 mm), and (iii) single-hit and multiple-hit low-velocity impact tests on skin panels (340 mm × 210 mm). The tensile and shear tests have already been thoroughly reported in a previous reference [29] and will not be discussed here.

Low-velocity impact tests were performed on coupon level samples (dimensions: 150 mm × 100 mm) according to ASTM D7136 [38]. Laminates with 24 plies (thickness: 4.8 mm) were manufactured with [45/0/−45/90]_3s_ layup and tested at impact energies of 8 J, 10 J, and 12 J with a drop weight equipment using a hemispherical impactor (diameter: 16 mm). The force–time histories were recorded using a load cell (Kistler 9331B, Winterthur, Switzerland), an amplifier (Kistler 5011B, Winterthur, Switzerland), and NI 9239 data acquisition system (National Instruments Corp, Austin, TX, USA). Besides, skin panels (dimensions: 400 mm × 270 mm × 3.42 mm) produced with 18 plies and a stacking sequence of [±45/0/±45/∓45/90/0]_S_ were impacted first at the center, with an energy of 35 J or 40 J, followed by off-central impacts, with an impact energy of 25 J. Each of these panels experienced up to two low-velocity impacts at different locations according to Figure 1.

## 3. Numerical Simulation

### 3.1. Material Constitutive and Damage Model

Puck inter-fiber failure criterion [39,40] for unidirectional composites was considered for the CFRP plies. Puck failure equations (Equations (3) and (4)) calculate the stress in tension ($\sigma_{n}\left(\theta \right)>0$) and compression ($\sigma_{n}\left(\theta \right)<0$), which are used to compute the index ranging from 0 to 1, for unloaded condition to failure initiation, respectively.
(1)$$FF1=\frac{\sigma_{11}}{R_{∥}^{t}}$$
(2)$$FF2=\frac{\sigma_{11}}{R_{∥}^{c}}$$
(3)$$IFF1\left(\theta \right)=\sqrt{{\left[\left(\frac{1}{R_{⊥}^{t}}-\frac{P_{⊥\Psi }^{t}}{R_{⊥\Psi }^{A}}\right)\sigma_{n}\right]}^{2}+{\left(\frac{\tau_{nt}}{R_{⊥⊥}^{A}}\right)}^{2}+{\left(\frac{\tau_{nl}}{R_{⊥∥}}\right)}^{2}}+\frac{P_{⊥\Psi }^{t}}{R_{⊥\Psi }^{A}}\sigma_{n}\;for\;\sigma_{n}\geq 0$$
(4)$$IFF2\left(\theta \right)=\sqrt{{\left(\frac{\tau_{nt}}{R_{⊥⊥}^{A}}\right)}^{2}+{\left(\frac{\tau_{nl}}{R_{⊥∥}}\right)}^{2}+{\left(\frac{P_{⊥\Psi }^{c}}{R_{⊥\Psi }^{A}}\sigma_{n}\right)}^{2}}+\frac{P_{⊥\Psi }^{c}}{R_{⊥\Psi }^{A}}\sigma_{n}\;for\;\sigma_{n}<0$$
where θ is the angle of rotation of the coordinate system around the $x_{1}$ axis, $\sigma_{n}\left(\theta \right),\tau_{nt}\left(\theta \right),\tau_{n1}\left(\theta \right)$ are the stresses on the action plane, and $R_{∥}^{t}$ and $R_{∥}^{c}$ are the material strength in the fiber direction in tension and compression, respectively. $R_{⊥}^{t}$ are $R_{⊥∥}$ are the material strengths in normal and shear directions, respectively. While:(5)$$\frac{P_{⊥\Psi }^{t}}{R_{⊥\Psi }^{A}}=\frac{P_{⊥⊥}^{t}}{R_{⊥⊥}^{A}}cos^{2}\Psi +\frac{P_{⊥∥}^{t}}{R_{⊥∥}}sin^{2}\Psi$$
(6)$$\frac{P_{⊥\Psi }^{c}}{R_{⊥\Psi }^{A}}=\frac{P_{⊥⊥}^{c}}{R_{⊥⊥}^{A}}cos^{2}\Psi +\frac{P_{⊥∥}^{c}}{R_{⊥∥}}sin^{2}\Psi$$
and:(7)$$cos^{2}\Psi =1-sin^{2}\Psi =\frac{\tau_{nt}^{2}}{\tau_{nt}^{2}+\tau_{nl}^{2}}$$
(8)$$R_{⊥⊥}^{A}=\frac{R_{⊥}^{c}}{2\left(1+P_{⊥⊥}^{c}\right)}$$
where $P_{⊥⊥}^{t}$, $P_{⊥∥}^{t}$, $P_{⊥⊥}^{c}$, and $P_{⊥∥}^{c}$ are the inclination parameters for the master fracture body.

As Puck’s IFF requires the determination of the fracture plane with the maximum exposure value, an algorithm should be included during the material implementation to search for that plane [29]. The Simple Parabolic Interpolation Search (SPIS), a non-iterative algorithm for the search of a fracture angle without the need for iterative methods, has been used for each element during the explicit simulations. The SPIS is described in detail in a previous publication of the group [41].

The non-linear behavior was considered here using a third-degree polynomial (Equation (9)) [4].
(9)$$\tau_{12}=c_{1}\gamma_{12}^{3}+sign\left(\gamma_{12}\right)\cdot c_{2}\gamma_{12}^{2}+c_{3}\gamma_{12}$$
where the coefficients $c_{1}$, $c_{2},$ and $c_{3}$ were obtained by fitting Equation (9) to a stress–strain curve from the in-plane shear test. Since the non-linear shear strain is irreversible, the total strain is separated into elastic and plastic strain [42]:(10)$$\gamma_{12}=\gamma_{12}^{e}+\gamma_{12}^{p}$$

The irreversible plastic strain, $\gamma_{12}^{p}$, is calculated as:(11)$$\gamma_{12}^{p}=\gamma_{12}-\frac{\tau_{12}}{G_{12}^{0}}$$
where $G_{12}^{0}$ is the initial shear modulus, and the unloading takes place according to $G_{12}^{0}$.

After initiation of fiber or inter-fiber damage, the material is degraded according to a strain-based post-damage law. The degradation is controlled by defining several damage variables based on strain softening. The stiffness parameters are degraded selectively, based on the failure mode that occurred. The relationship between the damage variables and the stiffness parameters is shown in the damage matrix (Equation (12)):(12)$$\begin{array}{cc} dC=\left[\begin{array}{cccccc} {\left(d_{f}E_{1}(1-d_{m}v_{23}v_{32})\right)}_{/△} & {\left(d_{f}E_{1}(d_{m}v_{2}1-d_{m}v_{31}v_{23})\right)}_{/△} & {\left(d_{f}E_{1}(v_{3}1-d_{m}v_{21}v_{32})\right)}_{/△} & 0 & 0 & 0\\ & {\left(d_{m}E_{2}(1-d_{f}v_{13}v_{31})\right)}_{/△} & {\left(d_{m}E_{2}(v_{3}2-d_{f}v_{12}v_{31})\right)}_{/△} & 0 & 0 & 0\\ & & {\left(E_{3}(1-d_{m}d_{f}v_{12}v_{21})\right)}_{/△} & 0 & 0 & 0\\ & & & d_{m}G_{12}^{0} & 0 & 0\\ & symmetric & & & d_{m}G_{13} & 0\\ & & & & & d_{m}G_{23} \end{array}\right]\\ \mathrm{for}\;\Delta =\left(1-d_{f}d_{m}v_{12}v_{21}-d_{m}v_{23}v_{32}-d_{f}v_{31}v_{13}-2\;d_{f}d_{m}v_{21}v_{32}v_{13}\right) & \end{array}$$
where $d_{f}$ and $d_{m}$ are the damage variables for the fiber and matrix, respectively, defined by Equation (13).
(13)$$\begin{array}{cc} d_{f}= & \left(1-d_{FF1}\right)\left(1-d_{FF2}\right)\\ d_{m}= & \left(1-d_{IFF1}\right)\left(1-smc\cdot d_{IFF2}\right) \end{array}$$
and smc is equal to 0.6, representing a lower limit for the matrix damage variable to avoid numerical instability and element distortion. $d_{FF1}$, $d_{FF2}$, $d_{IFF1}$, and $d_{IFF2}$ are the damage variables corresponding to the failure Equations (1)–(4). These variables follow the strain-softening Equation (14) and are greater than 0 when the corresponding failure equation is met.
(14)$$d_{i}=\frac{\varepsilon_{i,u}}{\varepsilon_{i,u}-\varepsilon_{i,on}}\left(1-\frac{\varepsilon_{i,on}}{\varepsilon_{i}}\right)$$
where the subindex i represents each of the failure modes; $\varepsilon_{i,u}$ is the ultimate failure strain, calculated by 2GiSil, and $G_{i}$ and $S_{i}$ are the fracture energy and failure stress for each mode, respectively; l is the element characteristic length calculated from the cube root of the element volume at the beginning of the simulation, included in the simulations according to the crack-band model [43] to avoid mesh sensitivity problem; $\varepsilon_{i}$ is the equivalent strain input from simulation, being equal to $\varepsilon_{1}$, $<-\varepsilon_{1}>$, $\sqrt{\varepsilon_{n}{}^{2}+\gamma_{nt}{}^{2}+\gamma_{nl}{}^{2}}$, $\sqrt{\varepsilon_{n}{}^{2}+\gamma_{nt}{}^{2}+\gamma_{nl}{}^{2}}$ for $d_{FF1}$, $d_{FF2}$, $d_{IFF1}$, and $d_{IFF2}$, respectively; $\varepsilon_{i,on}$ is the failure onset strain, calculated when the failure criterion is met.

### 3.2. Damage Model for the Interface

Here, the damage on the interface is based on the bi-linear traction-separation law which uses a quadradic stress failure criterion (Equation (15)) to predict damage initiation and the B-K mixed model fracture energy law (Equation (16)) [44] to predict delamination propagation.
(15)$$\frac{t_{n}{}^{2}}{N^{2}}+\frac{t_{s}^{2}}{S^{2}}+\frac{t_{t}^{2}}{S^{2}}=1$$
where <·> is the Macaulay bracket to only consider the contribution of tensile normal traction at the interface; and *N* and *S* are the normal and shear strengths of the interface, respectively.
(16)$$G_{n}^{C}+\left(G_{s}^{C}-G_{n}^{C}\right){\left\{\frac{G_{s}}{G_{T}}\right\}}^{\eta }=G^{C}$$
where $G_{n}^{C}$ and $G_{s}^{C}$ are the critical normal and shear fracture energies, respectively; η is the B-K material constant; and $G_{S}=G_{s}+G_{t}$ and $G_{T}=G_{n}+G_{S}$.

### 3.3. Finite Element Models

The finite element models were developed in four levels of complexity. The simplest simulations were performed on single elements subjected to tensile, compression, and shear cyclic loads, designed to verify the accuracy of the progressive damage model, particularly related to damage initiation, fracture energy, and degradation of properties. Tensile specimens were simulated to verify the interaction among different failure models and to enable direct comparison with experimental results. The results for these two levels have already been thoroughly reported in a previous reference [29] and will not be discussed here.

For the low-velocity impact model on coupon specimens of the 24-ply laminate (Figure 2a), a detailed layer-by-layer description of inter-laminar and intra-laminar damage and a comparison with experimental data is provided for the impacts at 8 J, 10 J, or 12 J. This is the same model used in [26].

The model was discretized with C3D8R (8-node linear brick with reduced integration) elements with enhanced hourglass control. The element size was defined as 1 mm × 1 mm at the impactor/laminate contact region, with a coarser mesh outside this region, following a mesh sensitivity analysis which found that suitable for this material and loading condition [26]. The interface between plies in the laminate was simulated using layers of the zero-thickness cohesive element (COH3D8) whose properties are compiled in Table 1.

Since the skin panels experienced up to two impacts at different locations, see Figure 1, two different impactors were defined at each impact location moving towards the panel with predefined initial velocities. The second impact was only applied after the debouncing of the previous impactor. The mechanical properties of the CFRP composites used in the simulation are presented in Table 1, and they were obtained either from experiments or from similar materials in the literature.

For the simulation of multiple impacts, illustrated in Figure 2b, the material state of the laminate at the end of each loading step becomes the initial material state for the next loading step. This should include element stress and strain state, displacement and velocity, and damage indicators for the plies and interfaces. An initial predefined field option of Abaqus/Explicit [47] was used, as in [26]. For multiple impacts at the same location, a multi-step approach is found in the literature [25,26], i.e., applying the low-velocity impact in one simulation and then moving the deformed geometry to a new simulation and applying the subsequent impact. However, for the simulation of multiple impacts at different locations, as in the current work, the panel was hit by the impactors at the same simulation step.

This approach is considered valid if enough time passes after each impact event to mitigate unwanted oscillation. To investigate that, the force–time curve obtained for the second impact using two different simulation schemes is compared in Figure 3, that is, the multi-step simulation approach of hitting the panel once in each step, and the single-step approach with all impacts in one step. Since the response of the panel from both models is identical, the rest step was considered unnecessary for the simulation of low-velocity impacts on CFRP composite, as previously reported in [26]. Based on that, the multiple impacts were simulated in a single-step explicit FE model.

### 3.4. Parametric Study

Different impact locations and impact energies were simulated. These are identifiable according to the notation where the first term represents the impact event type (S for single impact and D for double impact), the second the impact location according to Figure 1 (i.e., C, S_1_, S_2_, S_3_, S_4_), and the third term the impact energy (25 J, 35 J, 40 J). For instance, S-S_1_-25J means a single 25 J impact at the S_1_ location in Figure 1. For the double impact cases, the first impact always occurs at the center of the skin panel at 40 J and the subsequent impact happens at one of the four different off-central locations with three different impact energies. So, only the second impact differs in these cases, and the notation was used to represent that second impact. A summary of all the parametric study cases is presented in Table 2.

## 4. Results and Discussion

In this section, the experimental and numerical results of the coupon under a low-velocity impact and skin panels subjected to single and multiple impacts at different locations are discussed.

### 4.1. Verification of the FE Simulations

The experimental and numerical curves force–displacement curves for the 8 J and 12 J impacts are compared in Figure 4. The numerical model provided accurate predictions of peak force, absorbed energy, peak displacement, and impact time duration.

The absorbed energies for the CFRP laminates subjected to impact at different energies are compared in Table 3. The underestimation in energy absorption can be attributed to the underestimation in damage by the numerical model. The numerically observed delamination area was compared to the projected delamination area measured using the Ultrasonic C-scan technique in Table 3. The error in the delamination area was calculated as less than −7% for all impact energies.

Figure 5 shows the layer-by-layer matrix damage predicted by the progressive damage model with the Puck failure criterion. The layer-by-layer analysis of matrix damage shows that due to the presence of dominant shear stresses at the plies in the middle of the laminate, the prediction of the damaged area did not follow the fiber orientation at each layer. The bending deformation in thin laminates results in high tensile bending stress at the rear side of the specimen, which is the location of tensile matrix damage initiation [29]. The matrix tensile damage propagated in-plane and through the thickness of the laminate until the rebounding of the impactor and no further propagation was observed during the rebounding. On the other hand, the compressive matrix damage that initiates in the contact point of the impactor and the laminate propagates towards the rear side. The dominant out-of-plane shear stresses in the middle plies result in the misalignment of the damaged area with the fiber orientation in the plies. Different damage patterns at the top, middle, and bottom layers were also reported in [48]. Matrix compression damage was only found in the layers near the impact side and the area under the impactor tip, while matrix tensile damage was greater in the bottom layers due to the bending effect. A similar observation was reported by [49,50] for different failure criteria such as Puck and Hashin.

### 4.2. Low-Velocity Impact on Skin Panels

Figure 6a–i compares the experimental and numerical curves of force–time, force–displacement, and energy–time for the CFRP skin panels subjected to a single impact at 40 J or 35 J at the center, and to a second impact off-center at 25 J (after the first impact at 40 J at the center). The model was capable of accurately predicting the response of the panel subjected to impacts at the center or off-center. The values of impact time duration, impact peak force, maximum displacement, and absorbed energy were consistent with the corresponding experimental data. Additionally, the numerical model accurately predicted the drops in the force–time curves at different energies, attributed to the onset and propagation of the damage in the panel, with a detrimental effect on the mechanical properties, including stiffness. More fluctuations and force drop due to the presence of fiber failure are seen in comparison to the coupon specimens subjected to 8, 10, and 12 J impacts.

### 4.3. Effect of the Impact Location

A total of 15 low-velocity impact cases were considered, three of them at the center and the others off-central, i.e., at S_1_, S_2_, S_3_, or S_4_ (see Figure 1). Figure 7a,b compares the force–time and force–displacement curves of the skin panels subjected to a single 40 J impact and an increase in peak force and a decrease in impact time duration is observed when the impact location shifts from the center of the plate towards the boundary. Changing the impact location from S_3_ to S_4_, which is the closest to the plate’s boundary, led to a more pronounced change in the response. A significant increase in impact bending stiffness of the panel can be observed in Figure 7b when the impact location is closer to the panel boundaries, and the maximum displacement decreased in the off-center impacts being the lowest for location S_4_. The energy–time curve in Figure 7c shows that shifting the impact location towards the clamped edge increased total absorbed energy. This was also observed for the 25 J and 40 J impacts. A similar finding was reported by [35] for woven glass fiber-reinforced epoxy subjected to center and off-center impacts.

Figure 8a–d summarizes the response parameters for all locations. The ratio of absorbed energy to impact energy is higher for off-center locations, which is consistent with the increase in the projected delamination area (Figure 8d). This has also been observed in the experimental findings of [35]. Furthermore, for the 35 J impact, the absorbed energy for the central impact was 9.02 J, which increased by 49% to 14.09 J for the off-center impact at S_4_ location. Accordingly, a 33.7% increase in the delamination area was observed (Figure 8d).

### 4.4. Effect of Pre-Existing Impact Damage

Figure 9 compares the force–time, force–displacement, and energy–time curves between a pristine panel and a panel with pre-existing damage subjected to impact at 40 J at off-center locations. Figure 9a compares the force–time responses for impact at S_1_, showing significant changes in the impact response of the pristine specimen compared to the specimen with pre-existing damage. The higher peak force and maximum impact displacement for the repeated impact in Figure 9b were also reported in [26,37]. Figure 9c shows that the interference of the damaged area from the first and second impact significantly reduces the energy absorption of the specimen [26]. Similarly, the force–time/displacement response in Figure 9d,e for impacts at S_2_ shows higher peak force and maximum displacement for impact on the panel with pre-existing damage; however, it can be observed that the trends are mitigated compared to the impact at S_1_. Lower absorbed energy for repeated impact at S_2_ was observed in Figure 9f, which indicated the interference of damaged areas. The impact force–time, force–displacement, and energy–time responses at S_3_ are shown in Figure 9g–i, respectively. At this location, the pre-existing impact damage shows a less severe influence on the response. It can be observed from Figure 9j–l that for the impact at S_4_, the responses are identical, while for the impact at S_1_ the panel with pre-existing damage, showed higher peak force and displacement, and lower absorbed energy (see Figure 9a–h), as previously reported for CFRP panels subjected to repeated impacts [25,26].

The greater maximum displacement for double impacts due to the change in bending stiffness was discussed by [36], while here the bending stiffness reduction of the first impact did not influence the impact response for far-away impact locations, see Figure 10b. The low-velocity impact in CFRP materials can thus be considered to cause local damage. However, the extent of the local damage needs to be investigated since it depends on different factors, especially the impact energy.

To find the minimum distance at which the pre-existing damage becomes relevant, the responses to the impact at 25 J and 40 J of a pristine panel and a panel with pre-existing damage are compared in Figure 10. It can be observed that their mechanical response was very similar far from the center for both impact energies. The 25 J impact shows the same result also for S_2_ and S_3_ locations, while the pre-existing damage area was the same for all cases of this figure.

Figure 10c,d compares the energy absorption ratio and total delamination area for the panels subjected to a single hit and double hits. As before, for panels with pre-existing damage, a significant difference can be observed when the impact happens in the vicinity of the pre-existing damage. For instance, for the second impact at 40 J, the presence of the pre-existing damage led to a 22.4%, 20.1%, and 3.8% decrease in the absorbed energy compared to the pristine panel for impact locations S_1_, S_2_, and S_3_, respectively. This is consistent with the numerical results of [25,26] for repeated impacts at the same location.

The damage accumulation pattern shows a smaller increase from the first to the second impact for the S_1_ location, while for impacts further away from the pre-existing damage, the delaminated area at the end of the second impact nearly doubled compared to the first impact. This is due to the change in the energy absorption mechanism of the CFRP specimen with pre-existing damage. Since the damage is already present in the vicinity of the contact area, the panel deforms more, and the peak impact force increases to absorb the initial impact energy, which is later released back to the impactor as elastic deformation energy. Therefore, the energy absorption capacity of the panel is decreased [26].

A comparison of the propagation of damage for different repeated impact scenarios on skin panels is shown in Figure 11. In this case, the projected damaged areas of D-S_1_-40 J and D-S_4_-40 J models after the first (at center) and second (off-center) impacts are compared. It can be seen that for the first impact, the damaged region is similar in shape and area since in both models, the panel was subjected to a 40 J impact at the center. The damaged area for the impact at the center is compared with the c-scan measurement to demonstrate the accuracy of the model in predicting the damage. Significant changes in the damaged area can be observed after the second impact. Two district and non-interfering damaged regions were observed for the D-S_4_-40 J model, while in D-S_1_-40 J, there was interference between damaged regions resulting in a single region of damaged material. This was consistent with the results of Figure 9, where the second impact at S_1_ resulted in bigger displacements and lower energy absorption. It is also worth mentioning that for the second impact at S_4_, the damage propagated towards the boundaries of the panel instead of the pre-existing damaged region; this behavior was similar to the single impact at the same location, meaning the pre-existing damage area did not inflict a significant change in the damage propagation of the off-center impact.

## 5. Conclusions

In this paper, a continuum damage mechanics-based material model was used to investigate the mechanical response and damage in CFRP composites subjected to low-velocity impacts. The progressive damage model was implemented in Abaqus/Explicit and validated in different stages, from coupon specimens to skin panels subjected to low-velocity impacts, where the constitutive model based on the Puck failure criterion led to accurate results.

The analysis showed that the impact location has an important effect on the mechanical response and damage of composite skin panels due to the significant increase in impact bending stiffness when the impact location moves towards the panel boundaries. The presence of pre-existing impact damage of 805 mm^2^ from a 40 J impact at the center of the skin panel resulted in a more complex impact response of the panel that depended on the impact location with respect to the pre-existing damage area. For an impact far away from the previous impact, the panel shows no difference in response compared to a similar impact on the pristine specimen. For an impact in the vicinity of a pre-existing damaged area, there was a significant change in the response of the panel compared to that of a pristine specimen. For the former, greater impact peak force and displacement and lower energy absorption capacity were observed.

## Figures and Tables

**Figure 1 materials-16-00914-f001:**
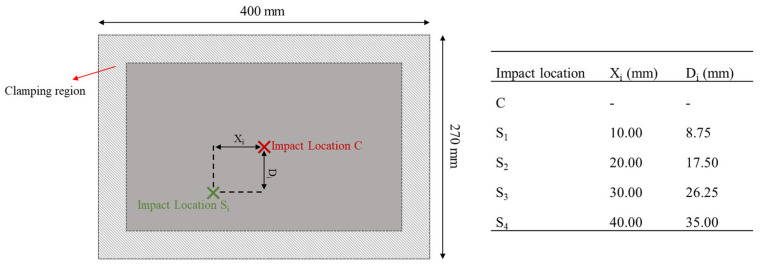
Impact locations on skin composite laminates.

**Figure 2 materials-16-00914-f002:**
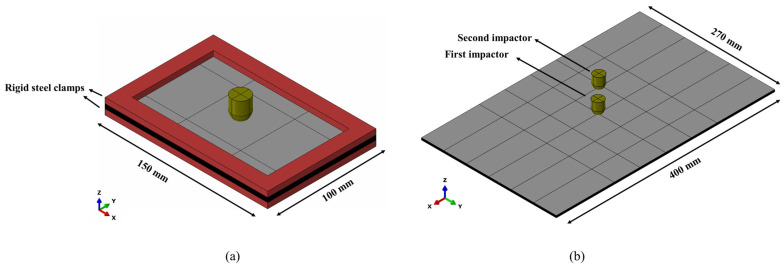
The FE models: (**a**) Impact on coupon specimen, (**b**) Multiple impacts on skin panel.

**Figure 3 materials-16-00914-f003:**
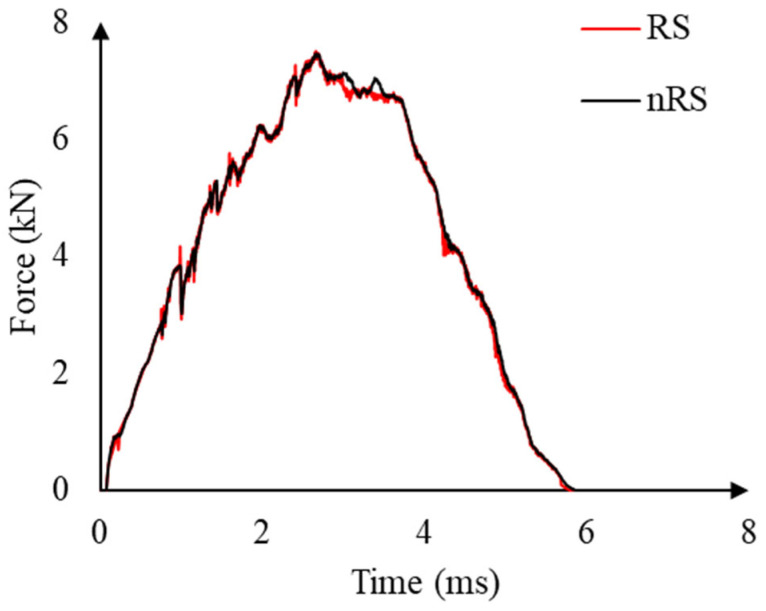
Comparison of force–time curves for the panel subjected to multiple impacts from simulations with the rest step (RS) and without the rest step (nRS).

**Figure 4 materials-16-00914-f004:**
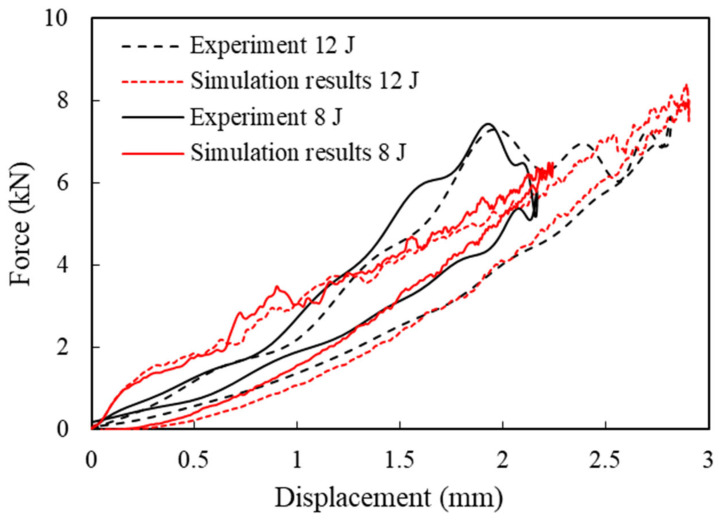
Comparison of experimental and numerical force–displacement curves for a low-velocity impact at 8 J and 12 J on a coupon specimen [29].

**Figure 5 materials-16-00914-f005:**
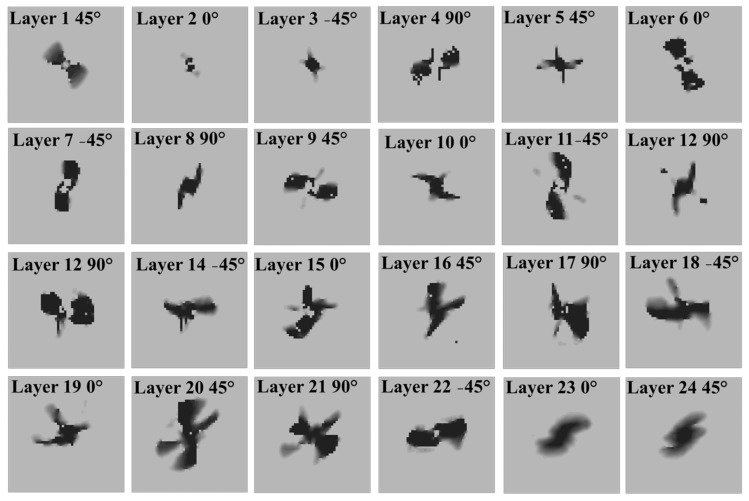
Layer-by-layer prediction of the matrix damage area, layer 1 being the impact side and layer 24 being the rear side of the specimen.

**Figure 6 materials-16-00914-f006:**
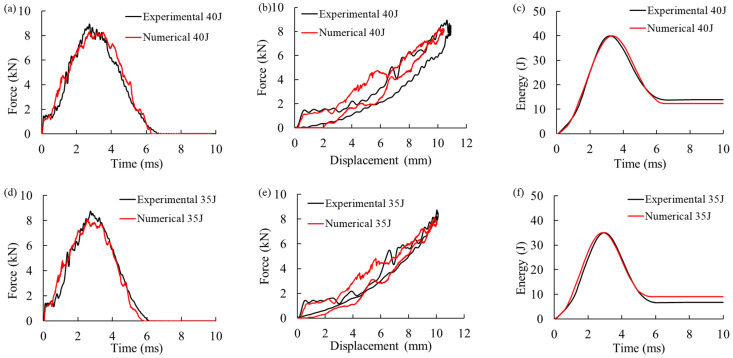
Experimental and numerical force–time, force–displacement, and energy–time curves for (**a**–**c**) Single impact at 40 J at the center, (**d**–**f**) Single impact at 35 J at the center, and (**g**–**i**) Second impact off-center at 25 J (after the first impact at 40 J at the center).

**Figure 7 materials-16-00914-f007:**
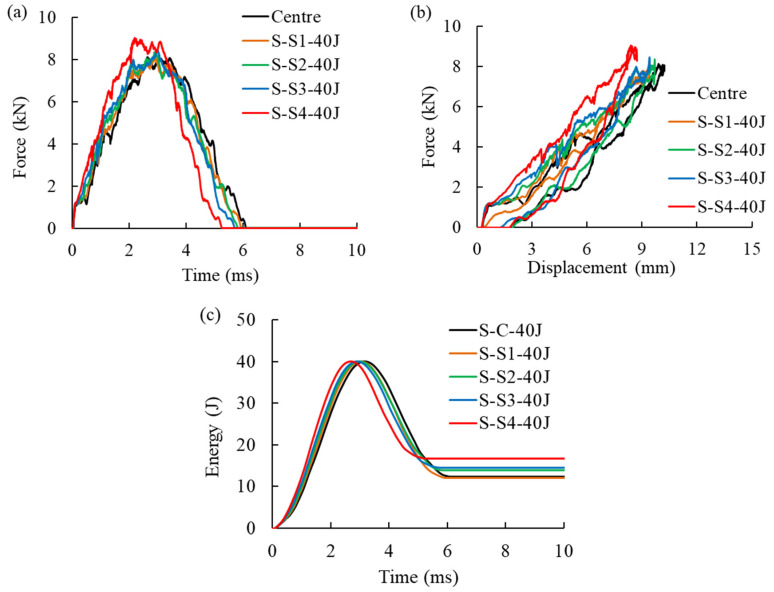
Comparison of force–time (**a**), displacement–time (**b**), and energy–time (**c**) curves for 40 J impact at the center and off-center locations.

**Figure 8 materials-16-00914-f008:**
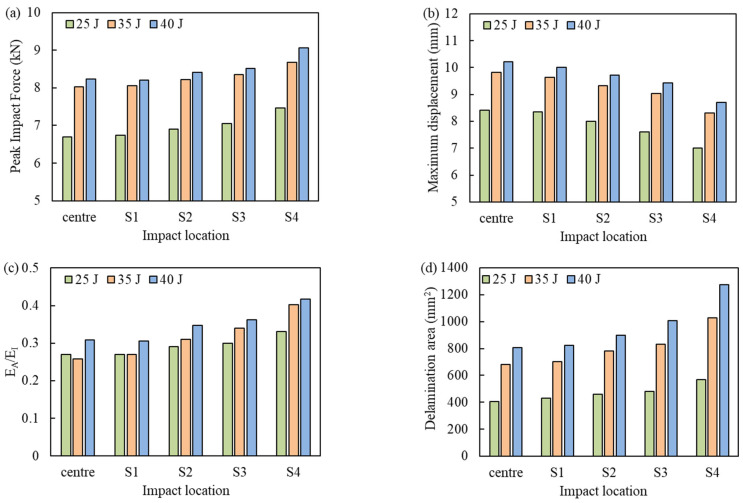
Comparison of the center and off-center low-velocity impact responses. (**a**) Peak impact force, (**b**) Maximum displacement, (**c**) Ratio of absorbed energy to impact energy, and (**d**) Projected delamination area.

**Figure 9 materials-16-00914-f009:**
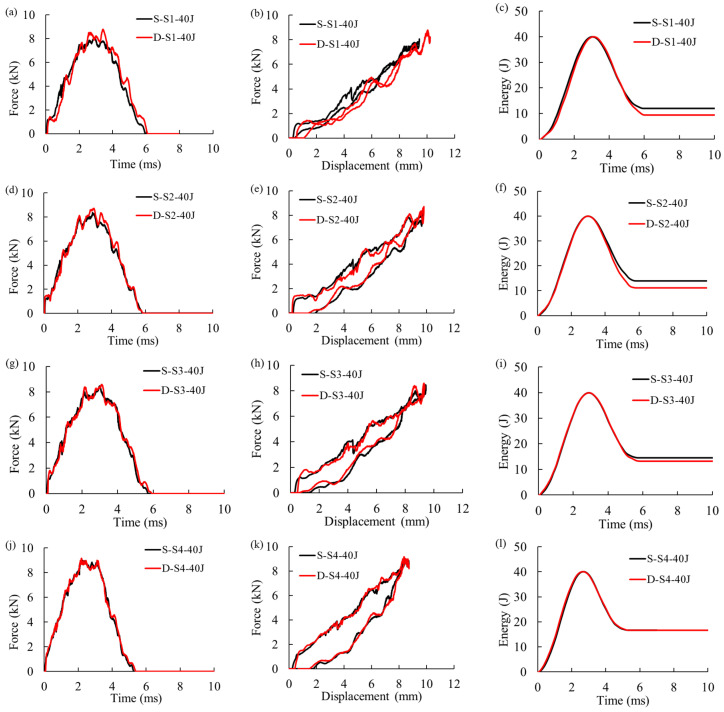
Comparison of force–time, displacement–time, and energy–time curves between a single and a double 40 J low-velocity impact at: (**a**–**c**) S_1_, (**d**–**f**) S_2_, (**g**–**i**) S_3_, and (**j**–**l**) S_4_.

**Figure 10 materials-16-00914-f010:**
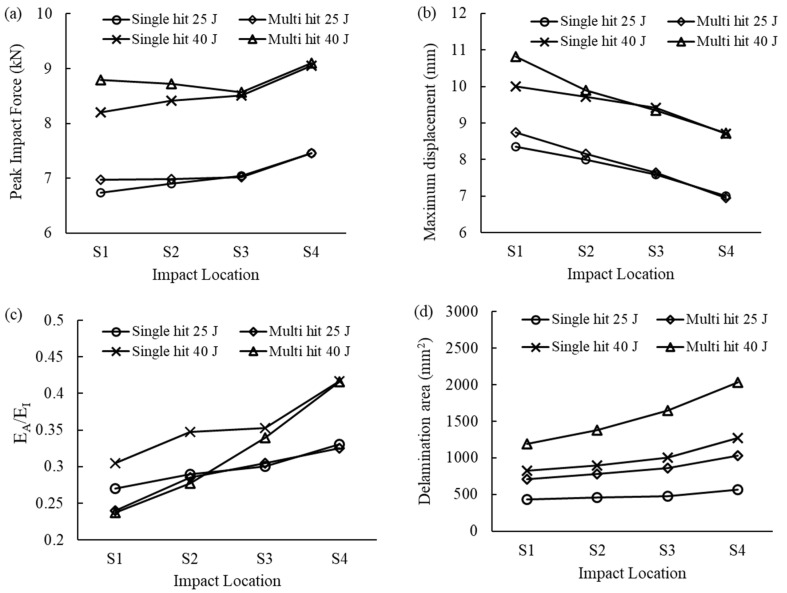
Comparison impact response for pristine panel and panel with pre-existing damage: (**a**) Peak impact force, (**b**) Maximum displacement, (**c**) Ratio of absorbed energy to impact energy, and (**d**) Sum of projected delamination area.

**Figure 11 materials-16-00914-f011:**
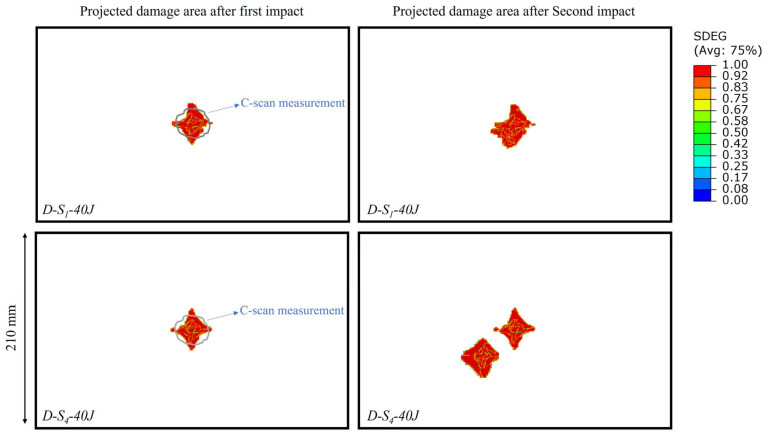
Comparison of the projected damaged area of D-S1-40 J and D-S4-40 J models after the first (at center) and second (off-center) impacts.

**Table 1 materials-16-00914-t001:** Mechanical properties of CFRP composite.

Laminae Properties		Interface Properties	
$E_{11}$ [GPa]	157.5 *	$E_{n}=E_{s}=E_{t}$ [GPa/mm]	5 ***
$E_{22}=E_{33}$ [GPa]	9.9 *	N [MPa]	33.0 ***
$G_{12}=G_{13}$ [GPa]	4.95 *	T [MPa]	54.0 ***
$G_{23}$ [GPa]	3.21 *	$G_{n}^{C}\;[N/\mathrm{mm}]$	0.6 ***
$c_{1},\;c_{2},\;c_{3}$ [MPa]	10.2 *, −0.5 *, 0.1 *	$G_{s}^{C}[N/\mathrm{mm}]$	2.1 ***
$v_{12}=v_{13}$	0.24 *		
$v_{23}$	0.35 **		
$R_{∥}^{t}$ [MPa]	2550.0 *		
$R_{∥}^{c}$ [MPa]	1350.0 **		
$R_{⊥}^{t}$ [MPa]	57.5 *		
$R_{⊥}^{c}$ [MPa]	199.8 **		
$R_{⊥∥}$ [MPa]	97.0 *		
$G_{ft},\;G_{fc}$ [N/mm]	133.0 **, 40.0 **		
$G_{mt},\;G_{mc}$ [N/mm]	0.6 **, 2.1 **		

* Data from test; ** Data from [4,45]; *** Data from [25,46].

**Table 2 materials-16-00914-t002:** Parametric study cases for different single low-velocity impact events at different locations, and for different multiple low-velocity impacts at different locations (for the double impacts, the notations only represent the results of the second impact since the first impact is equal to 40 J for all impact cases).

	Single Impacts at Different Locations		Double Impact at Different Locations	
Location	Impact Energy (J)	Code	Impact Energy (J)	Code
Center	25	S-C-25 J	-	-
	35	S-C-35 J	-	-
	40	S-C-40 J	-	-
S_1_	25	S-S_1_-25 J	25	D-S_1_-25 J
	35	S-S_1_-35 J	35	D-S_1_-35 J
	40	S-S_1_-40 J	40	D-S_1_-40 J
S_2_	25	S-S_2_-25 J	25	D-S_2_-25 J
	35	S-S_2_-35 J	35	D-S_2_-35 J
	40	S-S_2_-40 J	40	D-S_2_-40 J
S_3_	25	S-S_3_-25 J	25	D-S_3_-25 J
	35	S-S_3_-35 J	35	D-S_3_-35 J
	40	S-S_3_-40 J	40	D-S_3_-40 J
S_4_	25	S-S_4_-25 J	25	D-S_4_-25 J
	35	S-S_4_-35 J	35	D-S_4_-35 J
	40	S-S_4_-40 J	40	D-S_4_-40 J

**Table 3 materials-16-00914-t003:** Comparison of the experimental and numerical absorbed energies and projected delamination areas [29].

Impact Energy	Absorbed Energy			Delamination Area		
	Experimental (J)	Numerical		Experimental (mm^2^)	Numerical	
		Value (J)	Error (%)		Value (mm^2^)	Error (%)
8 J	3.41	2.98	−12.6	420	406	−3.3
10 J	4.43	3.82	−13.7	605	567	−6.3
12 J	4.88	4.57	−6.3	750	739	−1.5

## Data Availability

The data presented in this study are available on request from the corresponding author.

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
