# Peer review of "Experimental and Numerical Study of the Influence of Pre-Existing Impact Damage on the Low-Velocity Impact Response of CFRP Panels"

_materials, 2023, doi:10.3390/ma16030914_

Round 1
Reviewer 1 Report
The study of the low velocity impact response of composites, especially the effect of repeated impacts on the damage resistance is an interesting topic. The authors have designed their study well with combination of experiments and numerical models. The finite element simulation of the impact response of the Composites with Puck criteria is a valuable contribution. The manuscript can be accepted for publication with some minor changes.
My main concern is that while the location of the impact is shown to have a significant effect, the repeated impact has negligible effect on the force displacement curves. Perhaps the damage modes in individual plies have a strong effect but that has not been explored. The validation of the numerical model is done purely in terms of the force, displacement and energy curves. It would be better to include CT scan or ultrasonic C scan of the damage areas. Alternatively microscopic observation of the cross section of the impacted plates will give useful information to understand if the FE model has captured the different damage modes.
Since the validation of the numerical model allows a wider parametric study, the effect of multiple impacts (more than two) may also be explored for one case.
Author Response
The authors would like to acknowledge the editor and reviewers for their valuable comments, which helped us clarify some key points and improve the general quality of the manuscript. Our responses to the queries of the editor and reviewers are in the attached file. Changes are marked with the highlighted text in the revised manuscript.

Reviewer 2 Report
1. Further, please elaborate the figure 3, whether the identical state of the "Rest step" is necessary or unnecessary, else, please include the statement from reference 21, as you have mentioned it is taken from 21. For better clarity.
2. Give a numerical value in the conclusion of the pre-existing impact zone.
3. Figure 9 is the imperative section of the article, so please give a detailed explanation for every figure 9 (a to I), at least one line for each figure.
4. Layer-by-layer prediction is very nice but needs additional explanation.
5. Further inference can be given by comparing the experimental and numerical investigation.
6. Possible include a few relevant references from the years 2022 and 2021.
Author Response

(The authors gave the same response as above.)
